# Multinuclear Magnetic Resonance Study of Sodium Salts in Water Solutions

**Włodzimierz Makulski**

Faculty of Chemistry, University of Warsaw, Pasteura 1, 02-093 Warsaw, Poland; wmakul@chem.uw.edu.pl;
Tel.: +48-(22)-55-26-370

**Abstract:** The small amounts of gaseous $^3$He dissolved in low concentrated water solutions of NaCl, NaNO$_3$ and NaClO$_4$ were prepared and examined by $^3$He-, $^{23}$Na-, $^{35}$Cl- and $^{15}$N-NMR spectroscopy. This experimental data, along with new theoretical shielding factors, was used to measure the $^{23}$Na nuclear magnetic moment against that of helium-3 $\mu(^{23}$Na$) = +2.2174997(111)$ in nuclear magnetons. The standard relationship between NMR frequencies and nuclear magnetic moments of observed nuclei was used. The nuclear magnetic shielding factors of $^{23}$Na cation were verified against that of counter ions present in water solutions. Very good agreement between shielding constants $\sigma(^3$He$)$, $\sigma(^{23}$Na$^+)$, $\sigma(^{35}$Cl$^-)$, $\sigma(^{35}$ClO$_4^-)$, $\sigma(^{15}$NO$_3^-)$ in water at infinite dilution and nuclear magnetic moments was observed for all magnetic nuclei. It can be used as a reference nucleus for calculating a few other magnetic moments of different nuclei by the NMR method. An analysis of new and former $\mu(^{23}$Na$)$ experimental data obtained by the atomic beam magnetic resonance method (ABMR) and other NMR measurements shows good replicability of all specified results. The composition of sodium water complexes was discussed in terms of chemical equilibria and NMR shielding scale.

**Keywords:** $^{23}$Na nuclear magnetic moment; $^{23}$Na, $^{35}$Cl and $^{15}$N nuclear magnetic shielding; NaCl; NaClO$_4$; NaNO$_3$ water solutions; $^3$He reference nucleus

## 1. Introduction

Nuclear Magnetic Resonance (NMR) spectroscopy plays a crucial role in the experimental establishment of nuclear magnetic moments. Several of these experiments were performed in the gas phase in recent times [1,2]. Unfortunately, elemental sodium does not form any gaseous compounds. Some previous measurements of the dipole moment of $^{23}$Na from NMR spectra were performed in water solutions of sodium chloride [3,4]. The nuclear magnetic moment of sodium nucleus was also derived from the atomic beam magnetic resonance method [5]. Interestingly, both kinds of methods lead to slightly inconsistent results. Recent theoretical calculations of $^{23}$Na$^+$ shielding in water solutions do not generally resolve this discrepancy [6]. Therefore, we decided to explore water solutions of three ionic compounds that are very soluble in water: NaCl, Na$^{15}$NO$_3$ and NaClO$_4$. It is important that in each case we are able to measure NMR signals of the counterpart chlorine and nitrogen nuclei in ions present in the solutions: Cl$^-$, NO$_3^-$ and ClO$_4^-$. The good knowledge of absolute shielding scales of chlorine and nitrogen nuclei is of great significance here. It allows the absolute shielding constant of $^{23}$Na$^+$ cation known from theoretical calculations to be verified. And last but not least, the sodium magnetic moment was used previously as a nuclear reference for finding other nuclear moments using the NMR method. Improving this result seems to be important in this context.

$^{23}$Na is the only stable isotope of sodium. The other 20 isotopes, apart from $^{22}$Na (half-life 2.602 years) and $^{24}$Na (half-life 14.96 h), are all short living nuclei [7]. The magnetic properties for some of them are known to a certain extent. $^{23}$Na-NMR spectroscopy is a very attractive field of chemical research, mainly because of its biological and environmental importance. Sodium-23 has good NMR

receptivity, 0.0927 relative to [1]H-NMR spectra. Unfortunately, the relatively large quadrupole moment (Q = 0.1045 barn, 1 barn = $10^{-28}$ $m^2$) limits the scope of the scientific research because short relaxation times cause wide signals to occur and the collapse of its internal structure. Only in very symmetrical environments, where electric field gradients (EFG) are small, the observed resonance lines are relatively narrow. The linewidth of the [23]Na-NMR signal of reference compound—0.1M NaCl solution in $D_2O$—is ~8.2 Hz at room temperature. The moderate spectral range—72 ppm and resonance frequency near that of carbon spectra provide a good opportunity for completing many valuable research projects. [23]Na spectra were firstly used for investigations of sodium cation in different solvents and its mixtures according to concentrations and temperatures [8,9]. Solvation effects, complex formation and relaxation phenomena remain the central focus for many researchers.

Scalar values of nuclear magnetic moments are strictly connected to their shielding constants in the chemical species under study. So if they are well known, the proper use of this relationship can give precise shielding parameters of nuclei present in any chemical substance. The accurate recognition of this matter leads to reaching far beyond the limits of NMR spectroscopy and to the influencing of nuclear quantum physics.

## 2. Results

From the NMR spectroscopy point of view, the search for concentration behaviour of chemical shifts ($\delta$ in parts per million) for cations or anions is a common procedure in water solutions. These dependences cannot be simple (as in the gaseous state), particularly at high concentrations where specific ion pairs should be present. In small concentrated solutions the quadratic function is usually used where $\delta_0$ means the chemical shift at infinite dilution and higher coefficients $\delta_1$ and $\delta_2$ are responsible for changes in the magnetic susceptibility and intermolecular interactions effects of a particular ion with solvent molecules. The chemical shift is described by the frequency of the resonance line expressed against a standard compound which is defined as 0 ppm. This shifts can be converted to the absolute shielding scale when shielding a reference compound is well known. On the other hand, peak position can be directly read in the frequency scale expressed in MHz units. All these modes were utilised in this work. The aim of our work is to measure the sodium nuclear magnetic moment and compare it with previous results, in particular with (ABMR) atomic beam magnetic resonance findings. Measurement has been carried out relative to the nuclear magnetic moment of helium-3 dissolved in appropriate salt solutions.

Experimental NMR data measured in this work was collected in Table 1. The NMR radiofrequencies were precisely measured by spectrometer and they are strictly related to the form of ions structured in water layers around them. Solvation processes can change during movements through the solution when they undergo thermal motions. The simple relationship between sodium-23 and helium-3 frequencies was used to extract the [23]Na nuclear magnetic moment [1]:

$$\Delta\mu_{Na}^z = \frac{\nu_{Na}}{\nu_{He}} \cdot \frac{(1 - \sigma_{He})}{(1 - \sigma_{Na})} \cdot \frac{I_{Na}}{I_{He}} \Delta\mu_{He}^z \tag{1}$$

where $\nu_{Na}$ and $\nu_{He}$ are the appropriate radiofrequencies extrapolated to the infinite diluted water solutions of NaCl, $NaNO_3$ and $NaClO_4$. The $\Delta$ symbol and superscript "z" mean the projection of the full magnetic moment vector on the field axis $B_z$. $I_x$ is the magnetic quantum number of the measured nucleus, and $\sigma_{He,Na}$ are shielding corrections of nuclei in the experimental conditions. Analogously, another formula was used for the deuterium reference ("lock" system):

$$\Delta\mu_{Na}^z = \frac{\nu_{Na}}{\nu_D} \cdot \frac{(1 - \sigma_D)}{(1 - \sigma_{Na})} \cdot \frac{I_{Na}}{I_D} \Delta\mu_D^z \tag{2}$$

where $\nu_D$ and $\Delta\mu_D^z$ values belong to deuterium nuclei. Both the above equations lead to calculating the magnetic moment $\mu_{Na}$ when all other quantities are known. The appropriate concentration functions of $^{23}$Na extrapolations are shown in Figure 1.

**Table 1.** Nuclear Magnetic Resonance frequencies, chemical shifts and nuclear shielding parameters of $^{23}$Na, $^{35}$Cl, $^{15}$N, $^{3}$He and $^{2}$H*.

| Water Solutions | Nuclide | $\nu_0$ (Radiofrequency) MHz | $\delta_0$/ppm | $\delta_1$/ppm mL mol$^{-1}$ $\delta_2$/ppm mL mol$^{-2}$ | $\sigma$/ppm | Reference |
|---|---|---|---|---|---|---|
| NaCl | $(^{23}Na^+)_{aqua}$ | 132.4197693(5) | −0.045 | 0.0532 −0.0146 | 581.20 | [6] |
| | $^{35}Cl^-$ | 49.0491387(2) | 4.632 | 0.3928 −0.0748 | 998.36 | [This work] |
| | $^{3}He$ | 381.3564668(30) | −2.748 | −0.0522 | 59.709 | [This work] |
| NaNO$_3$ | $(^{23}Na^+)_{aqua}$ | 132.4197679(6) | −0.055 | −0.6459 0.1676 | 581.20 | [6] |
| | $^{15}NO_3^-$ | 50.7450281(5) | −5.589 | 0.132 −0.0715 | −137.31 | [This work] |
| | $^{3}He$ | 381.3564680(30) | −2.745 | −0.0161 | 59.706 | |
| NaClO$_4$ | $(^{23}Na^+)_{aqua}$ | 132.419768(7) | −0.055 | −0.8174 0.1914 | 581.20 | [6] |
| | $^{35}ClO_4^-$ | 49.0983353(4) | 1007.643 | −0.295 0.070 | −4.64 | [This work] |
| | $^{3}He$ | 381.3564673(35) | −2.747 | −0.0173 | 59.708 | |

* Lock system tuned to $\nu_0(D_2O)$ = 76.8464 MHz.

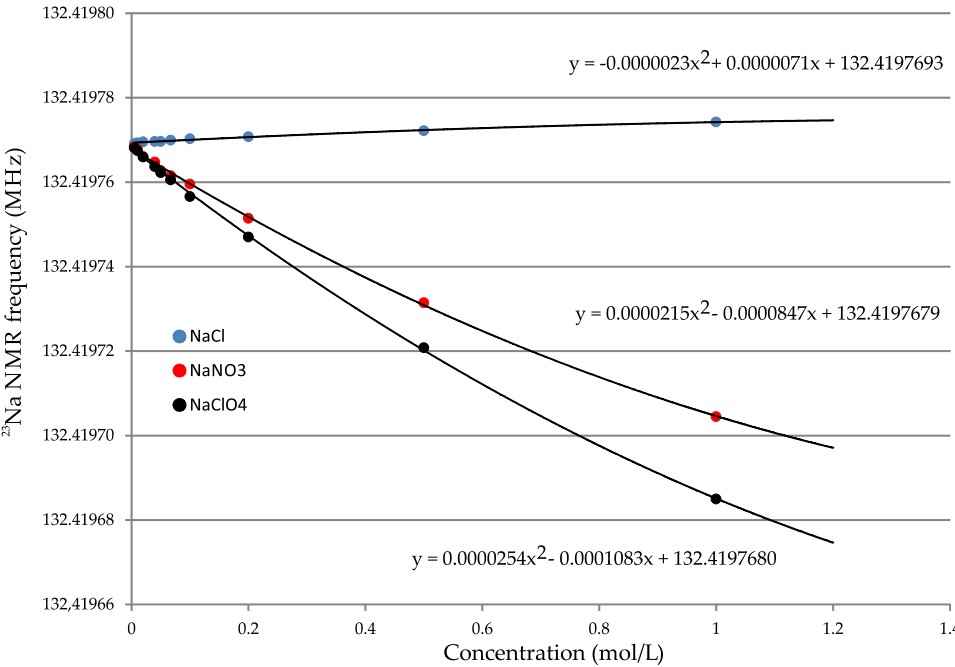

**Figure 1.** $^{23}$Na NMR frequencies recorded for sodium cation in aqueous solutions of NaCl, NaClO$_4$ and NaNO$_3$.

In general, the concentration dependences of chemical shifts/shielding for cations or anions cannot be linear, especially at concentrations above 1 M. Our analyses were done by single-variable quadratic functions and only $^{3}$He dependences show strict linearity. It is known that virial expansions can be used for the modelling of aqueous ionic solutions (see for example [10,11]. In principle the shielding parameter at infinite dilution, when the role of counterion can be omitted, should be modeled by

quantum chemical approaches. All coefficients of extrapolate relations are collected in Table 1 as $\delta_0$ [ppm], $\delta_1$ [ppm × mL × mol$^{-1}$] and $\delta_2$ [ppm × mL × mol$^{-2}$]. The course of the functions (Figures 1 and 2) reflects the macroscopic magnetic susceptibility effect of solutions and complex intermolecular forces occurred during rapid equilibration of solvent-separated cations and anions.

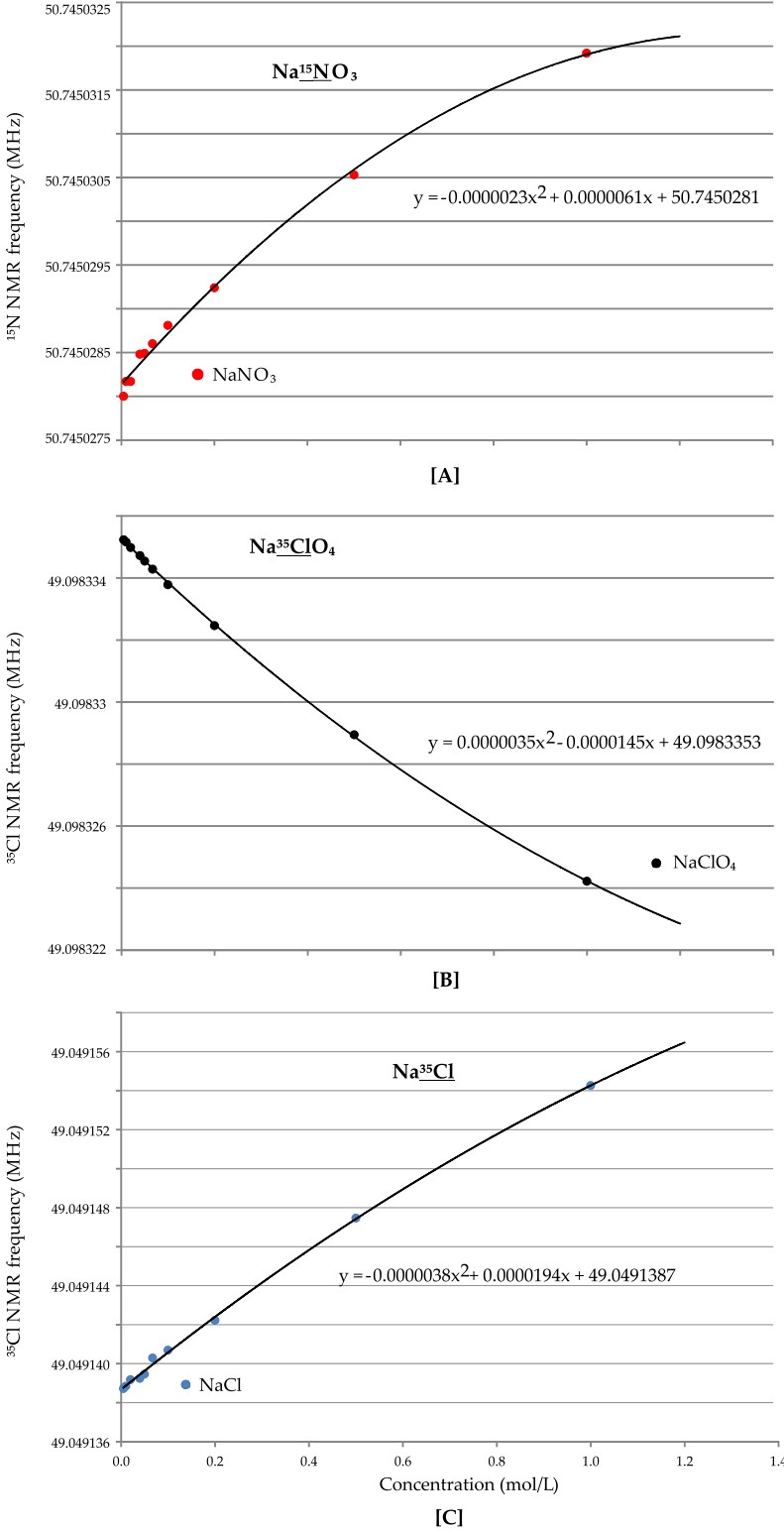

**Figure 2.** $^{15}$N and $^{35}$Cl NMR frequencies recorded for: **[A]** NO$_3^-$, **[B]** ClO$_4^-$, **[C]** Cl$^-$ ions in water solutions of NaNO$_3$, NaClO$_4$ and NaCl, respectively.

The newly measured $^{23}$Na nuclear magnetic moment can be checked and confirmed by a slightly reformulated Equation (3) when it is used in the form:

$$\sigma_{Na} = 1 - \frac{\nu_{Na}}{\nu_Y} \cdot \frac{\Delta\mu_Y}{\Delta\mu_{Na}} \cdot \frac{I_{Na}}{I_Y} \left(1 - \sigma_Y\right) \tag{3}$$

where $\sigma_Y$ means the shielding parameter for $^{15}$N and $^{35}$Cl nuclei in counterions present in the appropriate solutions: $NaNO_3$, $NaCl$ and $NaClO_4$. These calculations can give information as to how good shielding parameters of sodium cation in the water solution at infinite dilution are compared with theoretical results. They are shown in Table 2. The reference nuclei here are $^{35}$Cl and $^{15}$N. Shielding constants were taken from the experimental chemical shifts presented in Figure 2 and Table 2. The appropriate shielding values of reference substances, 0.1M NaCl and neat $CH_3NO_2$, were obtained from our previous investigations $\sigma_{Cl}$ = 1006(5) ppm and $\sigma_N$ = −135.8 ppm [2,12], respectively.

We can say, after all that hydrated cation is dynamically transformed between several structures:

$$Na(H_2O)_3{}^+ + H_2O \leftrightarrow Na(H_2O)_4{}^+ + H_2O \leftrightarrow Na(H_2O)_5{}^+ + H_2O \leftrightarrow Na(H_2O)_6{}^+ + H_2O \leftrightarrow Na(H_2O)_7{}^+ \tag{4}$$

Definitively, we choose molecular dynamics (MD) simulation results which suggest that the five coordinated sodium ion is the most populated structure (~60%) and the six-coordinated structure is the second most (~30%) [13]. These complexes were chosen as a starting point for detailed theoretical calculations of $^{23}$Na shielding constants of hydrated cations. The mean value of sodium ion shielding at the CCSD/utAdz level with relativistic correction (7.55 ppm) was established as 581.20 ppm with the uncertainty ±10 ppm. This value was taken by the authors for recalculating the sodium dipole moment from original experimental data [6]. For this work, shielding calculations were carried out using high level ab initio methods (CCSD) with correlation contributions and relativistic effects, while coordination numbers came from molecular dynamics (MD) investigations. Without a doubt, these results are the best known up to now in the relevant literature. In our work we prefer to consider the dynamic character of water surrounding the central cation by averaging results for 4, 5 and 6-coordination systems with its appropriate percentage. We finally obtained 580.12 ± 10 ppm for the shielding correction factor of sodium cations in liquid water.

The calculated results using Equations (1) and (2) are shown in Table 2. The $^{23}$Na nuclear moments from other sources: ABMR* experiment [5], recalculated from older NMR solution research [6] and tabulated values cited from NIST [14–16] are also included.

## 3. Discussion

### 3.1. ABMR Experiments with Sodium Atoms

The nuclear magnetic moment of $^{23}$Na nucleus was determined by the atomic beam magnetic method [17]. In this procedure the Larmor precession frequency ($\nu$) was measured in the known magnetic field (H). Both values are related with the nuclear magnetic moment by a simple formula: $\nu = \mu H / hI$, where I is a spin value of the given nucleus. A knowledge of spin numbers can provide a simple way to establish the magnetic moment. Only one resonance curve (resonance line) was observed for several different sodium compounds: $Na_2$, $NaF$, $Na_2B_4O_7$ and $NaCN$. The final magnetic moment of the $^{23}$Na nucleus was assumed as 2.216 nuclear magnetons $\mu_N$ ($\mu_N = e\hbar/2m_p$, where e is the elementary charge, and $m_p$ is the proton rest mass). Several improvements introduced much later in the ABMR method (separated oscillatory fields and triple resonance) provided an opportunity to measure the sodium moment with much better accuracy [5]. The authors measured the ratio $g_I/g_J$ = −0.40184406(40) for the $^{23}$Na atom in its ground state $2S_{1/2}$. Because $g_J$ is known as 2.0022960(7), the nuclear magnetic moment can be calculated as: $\mu(^{23}Na) = -g_I \, 3/2(m_p/m_e) \, \mu_N$. The original value was then assumed as 2.216082(2) $\mu_N$ without diamagnetic shielding correction and this was incorporated in the Table of Nuclear Magnetic Dipole and Electric Quadrupole Moments by Stone [14] as diamagnetically corrected.

We can slightly improve $\mu(^{23}Na)$ using the new, contemporarily used $m_p/m_e = 1836.15267389(17)$ [18] to give $\mu(^{23}Na) = 2.21608228\mu_N$ (without shielding corrections). This original value, was improved for the shielding effect in the sodium atom $\sigma(Na) = 640.62(500)$ ppm calculated by adding 60.5(10) ppm to the shielding of $Na^+$ cation according to the simple relation: $\sigma(atom)=\sigma(Na^+) + 60.5$ ppm (see Section 3.2 for details). This correction factor is in accordance with the theoretical result of $\sigma$ (Na, atom) = 637.1 ppm achieved in theoretical relativistic calculations and cited in Reference [19]. The final result was then stated as $\mu(^{23}Na) = 2.2175019(133)$ $\mu_N$ and marked as ABMR*. Both results were collected in Tables 2 and 3 for comparison and referred to in our later discussion.

**Table 2.** $^{23}Na$ nuclear magnetic moment and shielding constants measured in water solutions of sodium salts.

| $\mu(^{23}Na)/\mu_N$ | Method/System | Nucleus | $\sigma(^{23}Na)_{aqua}$ | Reference |
|---|---|---|---|---|
| 2.2175029(111) | NMR-$^3$He/NaCl | $^{35}Cl^-$ | 580.12 580.47 | [6] Theory Experiment [This work] |
| 2.2175029(111) | NMR-$^3$He/NaNO$_3$ | $^{15}NO_3^-$ | 575.37 | |
| 2.2175029(111) | NMR-$^3$He/NaClO$_4$ | $^{35}ClO4^-$ | 579.49 | |
| 2.2174964(111) | NMR-$^2$H/NaCl | $^{35}Cl^-$ | 577.55 | Experiment [This work] |
| 2.2174964(111) 2.2174964(111) | NMR-$^2$H/NaNO$_3$ NMR-$^2$H/NaClO$_4$ | $^{15}NO_3^-$ $^{35}ClO_4^-$ | 572.4 576.56 | |
| 2.2174997(111) | NMR-average | $^{35}Cl^-$ $^{35}ClO_4^-$ | 579.00 578.02 | [This work] |
| 2.2174982(233) | NMR | $^{35}Cl^-$ $^{35}ClO_4^-$ | 578.35 577.37 | [6] This work |
| 2.2175019(133) | ABMR* | $^{35}Cl^-$ $^{35}ClO_4^-$ | 580.02 579.03 | [5] This work |
| 2.217522(2) | ABMR | $^{35}Cl^-$ $^{35}ClO_4^-$ | 589.08 588.09 | [20] This work |
| 2.2176556(6) | NMR | $^{35}Cl^-$ $^{35}ClO_4^-$ | 649.29 648.30 | [20] This work |

ABMR*: New shielding value calculated in this work and original ABMR frequency value.

## 3.2. NMR Evaluation of $\mu(^{23}Na)$

The crucial role in the proper establishing of the $^{23}Na$ nuclear magnetic moment is played by the shielding correction factor of sodium cation in water solution. So, the structure of solvated $Na^+$ is of prime importance here. $Na^+$ cation is the most popular ionic species ever investigated in aqueous solutions. Hydration of sodium cation was the subject of many spectroscopic, physico-chemical and theoretical modelling studies [20–22]. Details of experimental difficulties of the solvation behavior are due to the essential flexibility of the hydration shells. The water molecules that are nearest to the central metal cation form the first hydration sphere with directly bonded water molecules. Sometimes, the second and third hydration spheres should be taken into account. The problem of first and second hydration cells in water solutions was the subject of many experimental and theoretical studies in the past [23,24]. The second hydration shell around the sodium ion is currently a matter of discussion [25]. All these structures are dynamic and undergo continuous changes [26]. At first we can distinguish between concentrated and diluted solutions. From our point of view, those diluted where several effects like the role of the counterion and ion pairing can be omitted, are the most important [27]. In this case the more probable hydration number was experimentally established and is 5 [28]. The coordination number was recently established as 5.5 ± 0.3 by a combination of theory, X-ray diffraction (XRD) and

extended x-ray absorption fine structure (EXAFS) [29]. Na-O distances in water complex are measured as 2.384 Å(XRD) and 2.370 Å (EXAFS) depending on the method. The application of the classical Na-O Lennard-Jones potential predicts this distance as 2.39 Å in excellent agreement with the experiment. The hydration of sodium cation was calculated from the first principles of the molecular dynamics (MD) method where the dependence of the self-diffusion coefficient and rotational correlation time of water depend on the $Na^+$ concentration. The average coordination number of sodium cation in neat water was stated (DFT/BLYP level) as 5.0. The average coordination number obtained from classical molecular dynamics, is in this case, 5.2 for $Na^+$ in water. Accordingly, the six coordination number is also acceptable from computer simulations, and there have been attempts at the fully optimising geometry of $[Na(H_2O)_6]^+$ [30]. Among five coordination sodium complexes the (4 + 1) with four water molecules directly coordinated to the metal cation and next through the hydrogen bonding, has the lowest energy on the potential energy surface. Among six coordination structures the (4 + 2) is of the lowest energy [31]. This slightly complicated behaviour was then confirmed by several research methods such as X-ray, neutron diffraction and Monte Carlo (MC) or molecular dynamics (MD) discussed in details in Reference [25].

$^{23}$Na nuclear magnetic moments calculated from Equations (1,2), for each pair of nuclei Na/$^3$He, Na/$^2$H, were stored in Table 2. $^3$He and $^2$H(D) shielding factors were referenced against gaseous $^3$He shielding of an isolated helium atom $\sigma_0(^3He) = 59.96743(10)$ ppm [32] and against shielding in heavy water used as lock frequency $\sigma(D_2O) = 28.837$ ppm [33], respectively (see Table 1 and Section 4 for details). All final results are internally concordant; however, those obtained from 2H(D) data are slightly underestimated. The arithmetic averaged value gives our final result for the dipole moment of sodium nuclei—$\mu(^{23}Na) = 2.2174997(111)$ $\mu_N$. It is obvious that the error in this result comes mainly from an error in the theoretical calculation of the shielding constant of the $^{23}Na^+$ ion in water solution and can only be reduced to ± 5 ppm. The difference between our result and that of the ABMR method is only $1.0 \times 10^{-4}$%. Our new result can be compared with analogous NMR measurements made previously in NaCl solutions in heavy water [3] when new shielding correction factors were used. The small solvent effect $H_2O$-$D_2O$ (−0.038 ppm) can certainly be omitted. The estimated frequency ratio $\nu(^{23}Na)/\nu(^2H) = 1.7231746(4)$ used in Equation (2), with the shielding correction shown above, gives a final result very close to our average value, namely—$\mu(^{23}Na) = 2.2174962(114)$ $\mu_N$.

The nuclear magnetic moment is essentially a vector property and is defined as a component in the one direction, usually denoted as the z direction. $\mu(X)$ means the length of projection on this axis. The full length of the magnetic moment vector can be calculated as $\mu(^{23}Na)^{length} = \mu(^{23}Na) \sqrt{I(I+1)}/I = 2.862780(30)$ $\mu_N$ for "bare" sodium nucleus.

The nuclear magnetic dipole moment is always proportional to the total angular momentum I and the proportionality constant is known as the $g_X$-factor; it is a dimensionless quantity that characterises the nuclear magnetic moment. The g-factor is 1.478333(15) in our case (see Table 3). Magnetic properties of given nuclei can be expressed as a gyromagnetic ratio ($\gamma_X$) also known as the magnetogyric ratio. This value is the ratio of the magnetic moment to its angular momentum $\gamma_X = g_X\mu_N/\hbar$, where $\mu_N$ is the nuclear magneton ($5.050783699(31) \times 10^{-27}$ J $\times$ T$^{-1}$). The gyromagnetic ratio for the sodium nucleus is $\gamma(^{23}Na) = 7.08035(7) \times 10^7$ in rad s$^{-1}$ T$^{-1}$. For NMR spectroscopy purposes it is often written in MHz/T units—1.126873(11), which can easily be recalculated to the Larmor frequency in any NMR spectrometer if only the magnetic field is known. The newly evaluated g-factor and $\gamma_X$ values of the $^{23}$Na nucleus and other nuclei used in this work can be seen in Table 3.

### 3.3. Nuclear Shielding from Aqueous Solutions

All $\sigma(^{23}Na)$ results calculated from Equation (3) are reported in the fourth column of Table 2. These quantities should be compared with theoretical $\sigma(^{23}Na) = 580.12$ ppm averaged over 4, 5 and 6 coordinated water complexes. We see that all results are in the error range ± 5 ppm, with one exception, which suggests the better accuracy than that assumed in Reference [6]. This error was taken as the main source of inaccuracy in our final magnetic moment result. Excellent agreement

with numbers calculated from [35]Cl-NMR measurements is stated. Small underestimated results are observed from [15]N-NMR measurements. This effect can be related to the undervalued [15]N absolute nuclear magnetic shielding scale, which was estimated using the nonrelativistic approach, in contrast with that of the [35]Cl scale. The small but valuable additional effect of the order ~1.5–2 ppm should move the entire scale to higher values and improve nitrogen results. The enhancement of [15]N shielding scale is, then, strongly recommended and can be achieved by NMR spectroscopic measurements in the gas phase of the $NH_3$ molecule (usual used reference). The relativistic amendments will be of prime importance here. Experimental research stages are in progress in our lab. On the other hand, the calculations using Equation (3) based on nuclear magnetic moments from Stone's tabulation [20] leads to completely missed values which differ from the expected value by 8 or more ppm. It has proved that new measurements and recalculations of old experiments were necessary to give compact and accurate results.

### 3.4. Theoretical Predictions of Nuclear Moments

It is not our intention to discuss the theoretical achievements of [23]Na nuclear magnetic moment calculations. We prefer to show the last research results in this field for crude comparison. The electromagnetic properties of sd shell nuclei were recently examined from the shell model using the first-principles methods [34]. The theoretical results in the medium similarity renormalisation group approach, ab initio coupled-cluster effective interaction scheme and phenomenological universal sd-B effective interaction method are as follows: $\mu$([23]Na) (IM-SRG) = + 1.972$\mu_N$, $\mu$([23]Na) (CCEI) = +1.887$\mu_N$ and $\mu$([23]Na) (USDB) = +2.098 $\mu_N$. The authors were in reasonable agreement with the experimental results. However we see that the discrepancy between theory and experiment is 5.85%, 8,75% and 5.39% in each case. In this context it is worth noting that the differences between pure experimental values shown in Table 3 do not exceed 0.00025%. This means that certainly up to now, the nuclear magnetic moments appear to belong to an experimental rather than theoretical account.

**Table 3.** Electromagnetic properties of sodium-23, chlorine-35, nitrogen-15, helium-3 and deuterium-2 nuclei used in this work.

| Nuclide | $I^\pi$ | Abundance % [7] | $\mu/\mu_N$ | Reference | $g_I$ | $\gamma_X \times 10^7$ | Q/Barn [14] |
|---|---|---|---|---|---|---|---|
| [23]Na | 3/2[+] | 100 | 2.2174997(111) | [This work] | 1.478333(7) | 7.08035(4) | 0.1045(10) |
| | | | 2.2175019(133) | ABMR* [5] | | | |
| | | | 2.217522(2) | ABMR [14] | | | |
| | | | 2.2176556(6) | NMR [14] | | | |
| [35]Cl | 3/2[+] | 75.78(4) | 0.821721(5) | [12] | 0.547814(3) | 2.62371(1) | 0.0850(11) |
| [15]N | 1/2[−] | 0.37 | −0.2830571(10) | [1,2] | 0.5661142(2) | 2.7113568(5) | |
| [3]He | 1/2[−] | 0.000134 | 2.127625308(25) | [32] | 4.2552506(1) | 20.3801695(2) | |
| [2]H(D) | 1[+] | 0.0156 | 0.8574382346(53) | [35] | 0.857438235(5) | 41.0662919(2) | 0.00286(2) |

A few atomic nuclei investigated in this work show electric properties known as nuclear electric quadrupole moments (I ≥ 1). They are admittedly not the subject of this work; however their full image is placed in Table 3. They are as follows: Q([2]H(D)) = 0.00286(2), Q([35]Cl) = 0.0850(11) and Q([23]Na) = 0.1045(10) (all values in barn units). Analogically to the nuclear magnetic moments these values can be established by combining nuclear quadrupole coupling constants (NQCC) with theoretical calculations of electric field gradients (EFG). Electric nuclear moments are important for nuclear structure theories. We see that the knowledge of both moments—magnetic and electric can be useful in many aspects of chemical spectroscopy and nuclear theories.

### 4. Materials and Methods

NaCl (sodium chloride, random crystals, 99.98%, Aldrich, Saint Louis, MO, USA), $NaClO_4$ (sodium perchlorate hydrate, 99.99%, Aldrich) and $Na^{15}NO_3$ (sodium nitrate, [15]N 98 atom %, Aldrich) were used to prepare water solutions at concentrations in the range 0.005–1.00 M. Glass samples of

~0.3 mL volume ($\varphi_{ext.}$ = 4 mm, $\varphi_{int.}$ = 3 mm, and 56 mm long) were filled with appropriate solutions, degassed using the freezing-pumping method by liquid nitrogen. The freeze-pump-thaw procedure was repeated three times to ensure the sample was free of air. Then small amounts of $^3$He were added from the lecture bottle (ChemGas, Boulogne, France, 99.9%)—70 Torr (1 Torr = 133.32 Pa) in each case—taking place above the frozen solution. Finally, the ready glass ampoule was sealed by a torch and set aside to defreeze. $^3$He gas pressure was controlled in the glass vacuum line and glass sample by a precision gas pressure gauge. The upper limit of helium concentration was established as $\leq 3.0 \times 10^{-3}$ mol/L depending on the solubility of helium in a given solution. These ampoules were than fitted into 5 mm o.d. NMR test tubes (Wilmad-Lab Glass Co., Vineland, NJ, USA) with liquid $D_2O$ (99.9 atom % D, Aldrich) in the annular space.

$^{23}$Na-, $^{35}$Cl- and $^{15}$N-NMR spectra were recorded on an INOVA 500 MHz spectrometer (Varian, Palo Alto, CA, USA) in an sw5 probe (switchable) operated at 132.419, 49.049 and 50.745 MHz. $^3$He-NMR spectra were measured on a unique homemade (helium) probe at 381.356 MHz against a single 0.1 atm gas sample with pure 3-helium. The samples of 0.1 M NaCl in $D_2O$ and neat $CH_3NO_2$ were used as references in $^{23}$Na-, $^{35}$Cl- and $^{15}$N-NMR spectra. The lock system was operated at 76.8464 MHz (at $B_0$ = 11.7574 T). All measurements were performed at a constant temperature of 300 K. A small solvent ($H_2O/D_2O$) isotope effect was observed in $^{23}$Na-NMR spectra which was equal to $-0.038$ ppm and a valuable effect in $^{35}$Cl NMR spectra +4.67 ppm in 0.1 M sodium chloride solutions.

The half-height resonance line widths noticed were as follows: $\Delta_{1/2}(^{23}$Na$) = 5.3 \div 7.5$ Hz with digital resolution (d.r.) = 0.15 Hz, $\Delta_{1/2}(^{35}$Cl$^-) = 8.0 \div 9.5$ Hz with d.r. = 0.15 Hz, $\Delta_{1/2}(^{35}$ClO$_4^-) = 1.35 \div 1.45$ with d.r. = 0.15 Hz, $\Delta_{1/2}(^{15}$NO$_3^-) = 0.30 \div 0.45$ with d.r. = 0.2 Hz and $\Delta_{1/2}(^3$He$) = 0.35 \div 1.15$ with d.r. = 0.2 Hz. Zero-filling procedure to 256k points and line broadening processing were performed before FT processing for higher quality of spectra.

The magnetic susceptibility effect for neat water (3.006 ppm) was calculated from the equation $\sigma_{1b} = -4/3\pi\chi_V$ and $\chi_V = \chi_M/M_p \, \rho$, where molar susceptibility $\chi_M = -12.97$, molar mass $M_p = 18.0002$ and density $\rho = 0.999865$ g/mL were applied.

A few important nuclear magnetic properties for all nuclei used in this work were collected in Table 3. As well as the nuclear magnetic moments expressed in nuclear magnetons, the corresponding $g_I$ and gyromagnetic factors were shown.

## 5. Conclusions and Outlook

New experimental results of $^{23}$Na-, $^3$He- and $^2$H(D)-NMR measured for different sodium salts—NaCl, NaClO$_4$ and NaNO$_3$—with dissolved $^3$He gas in water solutions provided an opportunity for an accurate calculation of the sodium nuclear magnetic moment. Recent theoretical results of the shielding parameter of sodium water complexes and helium dissolved in these solutions were used to get diamagnetic corrections. The final value $\mu(^{23}$Na$) = 2.2174997(111) \, \mu_N$ is in very good agreement with ABMR results for sodium atoms and those of NMR experiments in NaCl reported previously. The main source of estimated error of the magnetic moment arises from diamagnetic correction factors in both NMR and ABMR experiments. In the first case, an excellent understanding of dynamical formations of water complexes of sodium cation is of crucial importance. A short discussion of these complicated processes was included in this paper. Our results are of two-fold importance. Firstly, they can be a better reference result for other NMR experiments in this field. There are six nuclear magnetic moments measured by the NMR method against that of sodium-23. All can be recalculated against our new value. Secondly, they are good results to compare with purely theoretical calculations of the sodium dipole moment, which is a stringent test of nuclear physics nuclei models. Up to now, the theoretical achievements have been far from our expectations and cannot be known up to a very high degree of accuracy. The difference between theory and experiment remains ~5–8%. A full understanding of sodium magnetic dipole moment composition, as for most atomic nuclei in the periodic table, remains unknown in quantum physics. Further progress can be made when several subtle effects of nucleon interactions are incorporated. We hope that the new result of

$\mu(^{23}Na)$ will replace old values presented in several tabulations and catalogues of nuclear magnetic moments. Additionally, we find that the nuclear magnetic dipole moment of $^{15}N$ nucleus needs to be reinvestigated in new experiments. The shielding constants of the nitrogen reference molecule $NH_3$ should, in particular, be improved by relativistic effects which can also improve our results.

The new relativistic results for nitrogen shielding in $NH_3$ appeared during an internal review of this paper. Aucar used the Dirac equation and experimental geometry to obtain $^{15}N$ shielding in ammonia. The difference between former nonrelativistic and this new result is ~+2 ppm [36]. The new value shifts the entire absolute shielding scale in a positive direction. If we use this number to recalculate shielding of nitrate anion, the $\sigma(NO_3^-) = 577.00$ and 574.05 ppm occur-in a better agreement with theoretical calculations taken in this work as a benchmark. We hope that suitable experiments performed in the gas phase and indispensable calculations will be completed swiftly.

**Funding:** This research received no external funding.

**Acknowledgments:** The author is indepted to Karol Jackowski for reading the manuscript of this work and making valuable comments.

**Conflicts of Interest:** The author declare no conflict of interest.

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
