# Peer review of "Multinuclear Magnetic Resonance Study of Sodium Salts in Water Solutions"

_magnetochemistry, doi:10.3390/magnetochemistry5040068_

Round 1

Reviewer 1 Report

This is an interesting paper on multinuclear magnetic resonance study of sodium salts in water solutions. The nuclear magnetic shielding data of 23Na cation were verified against that of counter ions present in water solutions. Very good agreement between shielding constants σ(3He), σ(23Na+), σ(35Cl‒), σ(35ClO4‒), σ(15NO3‒) in water at infinite dilution and nuclear magnetic moments were calculated for all magnetic nuclei. I could recommend the manuscript for publication since it is well and carefully organized. Few minor revisions are indicated below:

Page 2, lines 46-47. The linewidth of the 23Na NMR signal of reference compound – 0.1M NaCl solution in D2O – is 8.2 Hz”.

Since the linewidth of quadrupolar nuclei is strongly dependent on temperature, this information should be indicated in the text.

Page 2, lines 66-67. This shifts can be recalculated to the absolute shielding scale when shielding a reference compound is well known.”.

The English text should be improved.

Page 3, lines 101-105. Details of experimental difficulties of the solvation behavior are due to the essential flexibility of the hydration shells. The water molecules that are nearest the central metal cation form the first hydration sphere with directly bonded water molecules. Sometimes, the second and third hydration spheres should be taken into account. The second hydration shell around the sodium ion is at present a matter of discussion”.

Equation 3. Na(H2O)3+ +H2O ↔ Na(H2O)4+ + H2O ↔ Na(H2O)5+ + H2O ↔ Na(H2O)6+ + H2O ↔ Na(H2O)7+”.

With respect to the problem of hydration cells, the author should give reference to a fundamental work of Merbach and collaborators (Lincoln S.F. et al., Adv. Inorg. Chem. 42 (1995) 1; Powel D.H. et al., Encyclopedia of Nuclear Magnetic Resonance, Wiley, Chichester, 1996, p. 2664; Frey U. et al., Dynamics and Solutions of Fluid Mixtures by NMR, Wiley, 1995, pp. 263–307; Cusanelli A. et al., Chimia 50 (1996) 618).

Reviewer 2 Report

In this work, the Author analyses water solutions of different sodium salts (NaCl, NaClO4, and NaNO3) by multinuclear NMR spectroscopy. The aim is to obtain an accurate estimation of the nuclear magnetic moment of 23Na from resonance frequency measurements. NMR data obtained as a function of the concentration are interpolated to estimate the value of 23Na chemical shift at infinite dilution. By using 2H or 3He as internal standard, magnetic moment values are derived through already published relationships. This method was extended also to nuclei present in the counterions, such as 35Cl and 15N, to validate the consistency of the calculations. Data is compared and discussed against previous measurements.

My opinion is that this manuscript is novel enough and of interest for the readership of Magnetochemistry, so it deserves to be published. However, I feel that some changes are required before this text can be accepted for publication.

- The Author reports trends for the 23Na radiofrequency as a function of the concentration for the different salts. The different salts have different concentration behaviour likely as a result of the formation of transient ion pairs with the relative counterions. This has been reported by Van Geet (J. Am. Chem. Soc. 1972, 94, 5578) and the Author should include this reference.

- In line 88-89, it is stated that “it is known that virial expansions can be used for modelling of aqueous ionic solutions.” I think the Author should reference previous work on this aspect here.

- In the experimental section, line 234, “densities” should be “concentrations”, as the author reports molarity units. Also, does “samples” mean “capillaries”? Also, it is not clear how the Author prepares the NMR solutions. After freeze-pump-thaw, how is He inserted in the sample? How is the amount of He controlled?

- The present form of the manuscript is quite difficult to read and the Author should consider reshuffling some sections. As an example, there is a quite extensive commentary about the hydration shell of water in the results section that should go in the discussion. Also, the new data on 15N and 35Cl is presented in the discussion only, while it should go in the results. The last part in the Materials and Methods (line 258 to 276) does not belong to this section

- I also find the writing style quite hard to follow. I feel that several sentences need to be reworded, as they are difficult to understand. For instance, there is a misuse of different words, such as “received” instead of “obtained”, “completed” instead of “reported” (line 215) or “permanent” instead of “continuous” (line 105). I believe that proofreading will be beneficial for the readability of this manuscript.
